# THE CONSENSUS GAME: LANGUAGE MODEL GENERATION VIA EQUILIBRIUM SEARCH

**Athul Paul Jacob**[*]
MIT

**Yikang Shen**
MIT-IBM AI Lab

**Gabriele Farina**
MIT

**Jacob Andreas**
MIT

## ABSTRACT

When applied to question answering and other text generation tasks, language models (LMs) may be queried *generatively* (by sampling answers from their output distribution) or *discriminatively* (by using them to score or rank a set of candidate outputs). These procedures sometimes yield very different predictions. How do we reconcile mutually incompatible scoring procedures to obtain coherent LM predictions? We introduce a new training-free, game-theoretic procedure for language model decoding. Our approach casts language model decoding as a regularized imperfect-information sequential signaling game—which we term the CONSENSUS GAME—in which a GENERATOR seeks to communicate an abstract correctness parameter using natural language sentences to a DISCRIMINATOR. We develop computational procedures for finding approximate equilibria of this game, resulting in a decoding algorithm we call EQUILIBRIUM-RANKING. Applied to a large number of tasks (including reading comprehension, commonsense reasoning, mathematical problem-solving, and dialog), EQUILIBRIUM-RANKING consistently, and sometimes substantially, improves performance over existing LM decoding procedures—on multiple benchmarks, we observe that applying EQUILIBRIUM-RANKING to LLaMA-7B outperforms the much larger LLaMA-65B and PaLM-540B models. These results highlight the promise of game-theoretic tools for addressing fundamental challenges of truthfulness and consistency in LMs.

## 1 INTRODUCTION

Current language models (LMs) perform quite well on some tasks involving generation or verification of factual assertions—including question answering, fact-checking, and even unconditional text generation. But they are far from perfectly reliable, and there is increasing evidence that LMs actually grow more prone to generating false but frequently repeated statements with increasing scale (McKenzie et al., 2023). Further complicating matters, LMs offer multiple affordances for solving factual generation tasks. They may be used both *generatively* (e.g. by asking for the most probable answer to a question) or *discriminatively* (e.g. by presenting a (question, answer) pair and asking whether the answer is acceptable) and, these two procedures do not always produce consistent results: generative procedures may fail when probability mass is spread across multiple contradicting answers (Wang et al., 2022; Mitchell et al., 2022), while discriminative procedures may fail due to miscalibration (Han et al., 2022; Chen et al., 2022) or subtle dependence on question wording (Jiang et al., 2020). Given these noisy and often-conflicting signals, how should we distill out an LM's best guess at the truth?

This paper presents an approach for reconciling generative and discriminative LM decoding procedures by formulating decoding as a signaling game (Lewis, 2008) that we call the CONSENSUS GAME. At a high level, this game features a GENERATOR agent that must communicate an abstract `correct` or `incorrect` value to a DISCRIMINATOR agent, but may only do so using a set of candidate natural language strings (Fig. 1). Intuitively, an effective *strategy* for this game (i.e. a joint policy) is one in which the GENERATOR and DISCRIMINATOR agree on the assignment of strings to correctness values. Given such a strategy, we may inspect it to identify candidates agreed by consensus to be `correct`.

Doing so requires solving a multi-step game with a complex (string-valued) action space. In recent years, *no-regret learning* algorithms have emerged as the preferred technique to compute effective

---

[*]Correspondence to: apjacob@mit.edu

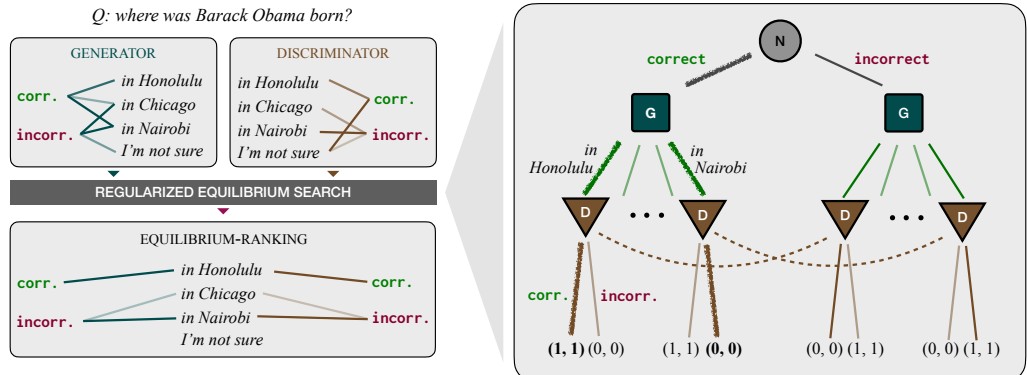

Figure 1: (Left) Overview of our approach. Differing LM queries fail to exhibit consensus about the answer to a factual question. By reconciling predictions between generative and discriminative LM queries using the CONSENSUS GAME, we obtain an accurate prediction. (Right) Structure of the CONSENSUS GAME, a two-player sequential signaling game with imperfect information. First, the environment (N) uniformly samples a correctness parameter. A GENERATOR (G) conditioned on this parameter produces a natural language string from a set of candidates. The DISCRIMINATOR (D) only observes this string and must predict the correctness parameter sampled by environment. If the DISCRIMINATOR correctly identifies this parameter, then both players receive a reward of 1. The dashed line connects nodes that are indistinguishable by the DISCRIMINATOR, since the DISCRIMINATOR does not observe the correctness parameter. By computing *regularized equilibrium strategies* for this game, we obtain predictions that reflect a consensus between the GENERATOR and DISCRIMINATOR.

strategies for such games, and have been successfully deployed in Poker (Brown & Sandholm, 2018; 2019), Stratego (Perolat et al., 2022), and Diplomacy (Bakhtin et al., 2023; FAIR et al., 2022; Jacob et al., 2022). Here, we show that they can also be applied to free-form language generation tasks. We call this game-theoretic approach to LM decoding EQUILIBRIUM-RANKING. Applied in 6 question answering benchmarks: MMLU (Hendrycks et al., 2020), ARC (Clark et al., 2018), RACE (Lai et al., 2017), HHH (Askell et al., 2021), TruthfulQA (Lin et al., 2022) and, GSM8K (Cobbe et al., 2021), EQUILIBRIUM-RANKING offers substantial improvements over existing generative, discriminative, and mixed decoding procedures. More generally, our results highlight the usefulness of the game-theoretic toolkit for formalizing and improving coherence in LMs. Improved coherence in turn leads to improved accuracy on factual tasks.

## 2 LANGUAGE MODEL CONSENSUS AS EQUILIBRIUM SEARCH

We study the problem of obtaining correct output from a **language model**, which maps input strings $x$ to output strings $y$ according to some distribution $P_{\mathsf{LM}}(y \mid x)$. While the techniques we present here are general, we focus in this paper on **question answering** problems consisting of a query $x$ (*In which of the following cities was Barack Obama born?*) and a set of candidate answers $\mathcal{Y}$ (*Honolulu, Chicago, . . .*) which may themselves have been sampled from the complete $P_{\mathsf{LM}}(\cdot \mid x)$. Given a set of candidates, we may use them with an LM in (at least) two ways:

- *Generatively*, by supplying as input (i) the query $x$, (ii) the set of candidates $\mathcal{Y}$, and (iii) a natural language prompt indicating that a correct answer is desired. In this case, the LM may be thought of as modeling a distribution $P_{\mathsf{LM}}(y \mid x, \texttt{correct})$, where the token $\texttt{correct}$ denotes the fact that the model was prompted to generate a correct answer.

- *Discriminatively*, by supplying as input (i) the query $x$ and (ii) a possible candidate answer $y \in \mathcal{Y}$, together with (iii) a prompt indicating that a correctness assessment $v \in \{\texttt{correct}, \texttt{incorrect}\}$ is sought. In this case, the language model acts as a model of as modeling a distribution $P_{\mathsf{LM}}(v \mid x, y)$ where $v \in \{\texttt{correct}, \texttt{incorrect}\}$.

These two approaches are conceptually equivalent. But as noted in the introduction, current LMs may give very different answers when queried in these different ways: answers produced generatively might be assessed `incorrect` with high probability or vice-versa. Research on LMs has proposed

two broad solutions to this problem. **Ensembling methods** (Ouyang et al., 2022; Li & Jurafsky, 2016; Glaese et al., 2022) simply combine discriminative and generative scores directly. While moderately effective, such approaches suffer from the fact that LM predictions are often poorly calibrated both within and across contexts, meaning that scores may not combine in meaningful or consistent ways. **Deliberation methods** (Wei et al., 2022; Yao et al., 2023; Du et al., 2023) perform this reconciliation within the LM itself, e.g. by re-prompting with competing inputs and an instruction to generate a textual justification for the best one. Such methods incur significant computational overhead.[1]

How might we design a principled and computationally efficient procedure for obtaining a "consensus" between competing LM predictions? Informally, a consensus prediction would satisfy two key properties: **coherence** (generative and discriminative scoring procedures should agree about which candidate answers are correct) and **reasonableness** (predictions should not be arbitrary, but instead as close as possible to original generator / discriminator behavior). The key idea in this paper is to operationalize these high-level desiderata in language of game theory, using **regularized equilibrium** concepts as formal framework for defining both coherence and reasonableness. Below, we introduce and explain this framework in detail, describing how to instantiate decoding as a signaling game, then compute equilibrium strategies of this game to obtain consensus LM predictions.

## 2.1 THE CONSENSUS GAME

Our approach to language generation begins by formulating language generation as a signaling game (Lewis, 2008) that we call the CONSENSUS GAME. The CONSENSUS GAME is played on a game tree, as depicted in Figure 1. At the start of the game (that is, at the root of the game tree), a *correctness parameter* $v \in \{\texttt{correct}, \texttt{incorrect}\}$ is selected uniformly at random by the environment. The correctness parameter is observed only by the GENERATOR, and controls whether the GENERATOR should aim to generate $\texttt{correct}$ or $\texttt{incorrect}$ answers. Upon observing this parameter, the GENERATOR produces a natural language string from a fixed set of candidates. Finally, this string is observed by the DISCRIMINATOR, who tries to guess the value of the correctness parameter by selecting one of $\{\texttt{correct}, \texttt{incorrect}\}$ as an answer. Both players obtain a **payoff** of 1 if the DISCRIMINATOR correctly identifies the value of the correctness parameter, 0 otherwise.

With this definition, it may be observed that players' expected **utilities** (the payoffs they may expect to receive) are as follows:

$$u_{\mathsf{G}}(\pi_{\mathsf{G}}, \pi_{\mathsf{D}}) := \frac{1}{2} \sum_{v \in \{\texttt{correct}, \texttt{incorrect}\}} \sum_{y \in \mathcal{Y}} \pi_{\mathsf{G}}(y \mid x, v) \cdot \pi_{\mathsf{D}}(v \mid x, y),$$

$$u_{\mathsf{D}}(\pi_{\mathsf{G}}, \pi_{\mathsf{D}}) := \frac{1}{2} \sum_{v \in \{\texttt{correct}, \texttt{incorrect}\}} \sum_{y \in \mathcal{Y}} \pi_{\mathsf{G}}(y \mid x, v) \cdot \pi_{\mathsf{D}}(v \mid x, y).$$

What is an effective strategy for maximizing these utilities? A standard answer to this question in the game theory literature is that a **Nash equilibrium** of the game should be sought. A Nash equilibrium is a pair of policies—one for the GENERATOR and one for the DISCRIMINATOR—such that each policy is optimal. That is, each player's strategy maximizes their expected given the other player's strategy. At a Nash equilibrium, no player has an incentive to unilaterally behave in any other way. In signaling games, Nash equilibria offer a natural way of formalizing the **coherence** criterion above: at equilibrium, both the GENERATOR and DISCRIMINATOR must agree about which messages correspond to $\texttt{correct}$ and $\texttt{incorrect}$ respectively in order to obtain a nonzero payoff.

However, Nash equilbria of the CONSENSUS GAME are not guaranteed to provide the second criterion of **reasonableness**. This is because the CONSENSUS GAME admits a multitude of Nash equilibria that are incompatible with the common-sense notion of truthfulness. For example, the strategy in which the GENERATOR deterministically maps $\texttt{correct} \mapsto$ "Nairobi", $\texttt{incorrect} \mapsto$ "Honolulu", and the DISCRIMINATOR maps "Nairobi" $\mapsto \texttt{correct}$, "Honolulu" $\mapsto \texttt{incorrect}$ forms a Nash equilibrium.

In order to sidestep the inappropriate equilibria and ensure reasonableness, we introduce a **regularization term** in the utility of the players, so that both the GENERATOR and the DISCRIMINATOR are penalized for settling on strategies that are far from some pair of *initial policies*: $\pi_{\mathsf{G}}^{(1)}$ and $\pi_{\mathsf{D}}^{(1)}$. By parameterizing these policies using a pre-trained LM, we may use knowledge about what answers

---

[1] As shown in Section 3, they are also orthogonal to, and composable with, the approach we propose here.

are likely to be correct *a priori* to guide selection of an equilibrium. As in Jacob et al. (2022), we incorporate this regularization term directly into the utility function (payoff) that the GENERATOR and DISCRIMINATOR attempt to optimize. Rather than the simple 0–1 payoff determined by agreement on the correctness parameter, they now attempt to optimize:

$$u_{\mathsf{G}}(\pi_{\mathsf{G}}, \pi_{\mathsf{D}}) := -\lambda_{\mathsf{G}} \cdot \mathrm{D_{KL}}[\pi_{\mathsf{G}}(\cdot \mid x, v) \parallel \pi_{\mathsf{G}}^{(1)}(\cdot \mid x, v)] + \frac{1}{2} \sum_{v} \sum_{y \in \mathcal{Y}} \pi_{\mathsf{G}}(y \mid x, v) \cdot \pi_{\mathsf{D}}(v \mid x, y),$$

$$u_{\mathsf{D}}(\pi_{\mathsf{G}}, \pi_{\mathsf{D}}) := -\lambda_{\mathsf{D}} \cdot \mathrm{D_{KL}}[\pi_{\mathsf{D}}(\cdot \mid x, y) \parallel \pi_{\mathsf{D}}^{(1)}(\cdot \mid x, y)] + \frac{1}{2} \sum_{v} \sum_{y \in \mathcal{Y}} \pi_{\mathsf{G}}(y \mid x, v) \cdot \pi_{\mathsf{D}}(v \mid x, y).$$

Note that the initial policies $\pi_{\mathsf{G}}^{(1)}(y \mid x, v)$ and $\pi_{L}^{(1)}(v \mid x, y)$ may be derived from an LM *prompted* with some initial string $x$, in order to obtain context-predictions (e.g. answers to a question). With these utilities, Nash equilibria for the game are pulled by the initial GENERATOR and DISCRIMINATOR policies in the direction of increased consensus.

Bakhtin et al. (2023) and FAIR et al. (2022) employed a similar regularization method for choosing actions, rather than messages, in versions of the board game Diplomacy. Franke (2013; 2017) have explored signaling games in the context of linguistic pragmatics to explain human language use. To the best of our knowledge, however, this is the first proposal for using regularized equilibrium concepts in signaling games to define target behavior in a language generation task. Additional related work is discussed in Appendix C.

## 2.2 EQUILIBRIUM-RANKING: LM RANKING VIA EQUILIBRIUM SEARCH

With this formulation, text generation requires finding a Nash equilibrium of the game with the utilities given above. How should we compute such an equilibrium? No-regret learning algorithms have emerged in recent years as the preferred technique to approximate equilibria in large games, and have been successfully employed to solve games at human or even superhuman level. At a high level, these algorithms find equilibrium by repeatedly interacting in the game and refining their policies after each iteration $t$. so as to minimize **regret** (the gap between the chosen action and the best action in hindsight).

In this section, we describe in detail how to perform no-regret learning in the CONSENSUS GAME in order to obtain consensus policies. Importantly, this approach modifies only signalling policies, and not the base policies $\pi_{\mathsf{G}}^{(1)}$ and $\pi_{\mathsf{D}}^{(1)}$ (i.e. the LM). In this sense, generating text by performing no-regret learning in the CONSENSUS GAME might be described as a *training-free consensus-planning method*. We call this method EQUILIBRIUM-RANKING.

**Initial policies** At time $t = 1$, that is, before any equilibrium computation has happened, EQUILIBRIUM-RANKING defines the initial policies $\pi_{\mathsf{G}}^{(1)}$ and $\pi_{\mathsf{D}}^{(1)}$ of the GENERATOR and DISCRIMINATOR, respectively, as follows. $\pi_{\mathsf{G}}^{(1)}$ normalizes $P_{\mathsf{LM}}^2$ across $v$ and $y$:

$$\pi_{\mathsf{G}}^{(1)}(y \mid x, v) \propto \frac{P_{\mathsf{LM}}(y \mid x, v)}{\sum_{v'} P_{\mathsf{LM}}(y \mid x, v')}.$$

Similarly for the DISCRIMINATOR, the initial policy normalizes across $y$ and $v$:

$$\pi_{\mathsf{D}}^{(1)}(v \mid x, y) \propto \frac{P_{\mathsf{LM}}(v \mid x, y)}{\sum_{y'} P_{\mathsf{LM}}(v \mid x, y')}.$$

This crucial step enables us to extract a well calibrated GENERATOR and DISCRIMINATOR from $P_{\mathsf{LM}}$. The specific form of the GENERATOR incorporates $v = \texttt{incorrect}$, and this is therefore a form of *self-contrastive* decoding (See, Section 3 for more details). This DISCRIMINATOR resembles approaches that query the LM itself to produce critiques (Ganguli et al., 2023; Chen et al., 2023b; Yao et al., 2023). However, to the best of our knowledge, this specific instantiation has not been explored in the past.

---

[2]In ARC, RACE, HHH, TruthfulQA, and GSM8K, based on prior work (Touvron et al., 2023; Brown et al., 2020), we additionally normalize $P_{\mathsf{LM}}(u|x)$ by the likelihood of the completion given "Answer:" as context: $P_{\mathsf{LM}}(u \mid \text{"Answer:"})$.

**Evolution of policies**    A classic observation in the theory of imperfect-information sequential games is that minimization of regret (viewed as a function of their overall policy on the game tree) can be achieved by solving separate, *local*, regret minimization problems at each information set (*i.e.*, decision point) of the game. This observation underpins the CFR framework (Zinkevich et al., 2007), as well as its generalization to more general convex losses, known as laminar regret decomposition (Farina et al., 2019). In our case, these techniques enable us to decompose the policy update of the players into separate updates for each correctness parameter $v$ (for the GENERATOR) and for each sequence $y$ (for the DISCRIMINATOR). We provide more detail and background in Appendix A.

In our setting, after operating the regret decomposition step, we find that the local regret minimization problems are composed of a bilinear term, plus a strongly convex KL-regularization term. Such composite utilities can be handled by the piKL algorithm (Jacob et al., 2022), which is specifically designed to perform regret minimization on KL-regularized objectives. In our setting, piKL prescribes that each player keep track of their average values:

$$Q_{\mathsf{G}}^{(t)}(y \mid x, v) := \frac{1}{2t}\sum_{\tau=1}^{t}\pi_{\mathsf{D}}^{(\tau)}(v \mid x, y), \qquad Q_{\mathsf{D}}^{(t)}(v \mid x, y) := \frac{1}{2t}\sum_{\tau=1}^{t}\pi_{\mathsf{G}}^{(\tau)}(y \mid x, v).$$

Each player then updates their policy according to:

$$\pi_{\mathsf{G}}^{(t+1)}(y \mid x, v) \propto \exp\left\{\frac{Q_{\mathsf{G}}^{(t)}(y \mid x, v) + \lambda_{\mathsf{G}}\log\pi_{\mathsf{G}}^{(1)}(y \mid x, v)}{1/(\eta_{\mathsf{G}}t) + \lambda_{\mathsf{G}}}\right\}, \qquad (1)$$

$$\pi_{\mathsf{D}}^{(t+1)}(v \mid x, y) \propto \exp\left\{\frac{Q_{\mathsf{D}}^{(t)}(v \mid x, y) + \lambda_{\mathsf{D}}\log\pi_{\mathsf{D}}^{(1)}(v \mid x, y)}{1/(\eta_{\mathsf{D}}t) + \lambda_{\mathsf{D}}}\right\}, \qquad (2)$$

where $\eta_{\mathsf{G}}, \eta_{\mathsf{D}} > 0$ are *learning rate* hyperparameters. piKL no-regret dynamics are known to have strong guarantees, including the following (more formal statements about the guarantees are available in Appendix A):

- **Convergence to an equilibrium point**. The average correlated distribution of play of GENERATOR and DISCRIMINATOR converges to the set of (regularized) coarse-correlated equilibria of the game.
- **Regularization toward reasonableness**. The average policy of any player remains within a radius of size roughly $1/\lambda_i$ from the initial policy $\pi_i^{(1)}$, where $\lambda_i$ is the amount of regularization applied to any player $i \in \{\text{GENERATOR}, \text{DISCRIMINATOR}\}$ (see Proposition 3).
- **Avoidance of regret**. The cumulative regret incurred by each of the players grows only logarithmic in the number of training steps (see Proposition 1).

At convergence, EQUILIBRIUM-RANKING returns $\pi_{\mathsf{G}}^*$ and $\pi_{\mathsf{D}}^*$, which are the refined GENERATOR and DISCRIMINATOR. While we do not provide a formal guarantee of convergence, we remark that the CONSENSUS GAME is an instance of a *potential* game (Monderer & Shapley, 1996), for which it is generally understood that decentralized no-regret learning dynamics similar to piKL converge in iterates to equilibrium (Anagnostides et al., 2022). Indeed, we witness good convergence properties in practice even without any perturbation. As mentioned earlier, convergence to a regularized Nash equilibrium is important to guarantee both coherence and reasonableness. Extensive empirical validation presented in the next section demonstrates the benefits of this approach in practice.

**Computational cost of our method.**    At each iteration, our method needs to update the policies $Q_{\mathsf{G}}^{(t)}, Q_{\mathsf{D}}^{(t)}$ according to (1) and (2). The number of operations at each iteration of the method is therefore linear in the number $|\mathcal{Y}|$ of sequences available to the GENERATOR.

## 3    EXPERIMENTS

As discussed in the previous section, EQUILIBRIUM-RANKING focuses on improving the *correctness* of language models in question-answering (QA) tasks. However, correctness manifests in various forms across different domains, including truthfulness, factuality, valid reasoning, value alignment, among others. Therefore, we will evaluate its performance on a diverse set of QA tasks: MMLU (Hendrycks et al., 2020), ARC (Clark et al., 2018), RACE (Lai et al., 2017), HHH (Askell et al.,

2021), and TruthfulQA (Lin et al., 2022). It's important to note that EQUILIBRIUM-RANKING is a sampling strategy and not a deliberation method like chain-of-thought (CoT) (Wei et al., 2022) and self-consistency (Wang et al., 2022). Nevertheless, we will demonstrate in GSM8K (Cobbe et al., 2021) that we can achieve some additional gains when combining EQUILIBRIUM-RANKING with self-consistency and CoT.

**Hyperparameters**  EQUILIBRIUM-RANKING has four parameters, $\eta_\mathsf{D}, \lambda_\mathsf{D}$ and $\eta_\mathsf{G}, \lambda_\mathsf{G}$. Although tuning these parameters will lead to better performance, in all our experiments we set $\eta_\mathsf{D} = \lambda_\mathsf{D} = \eta_\mathsf{G} = \lambda_\mathsf{G} = 0.1$. We run EQUILIBRIUM-RANKING for 5000 iterations [3]

**Actions in the CONSENSUS GAME**  As mentioned in Section 2, in order to make our approach amenable to current computational techniques, we make the modeling assumption that the GENERATOR picks distribution over a finite set of candidates $\mathcal{Y}$. In multiple-choices tasks, these are the multiple choice options. In generative tasks, a common approach to generate the finite set of candidates is via sampling with nucleus (Holtzman et al., 2019) and top-$k$ (Fan et al., 2018b) from the distribution $P_\mathsf{LM}(y \mid q, \mathtt{correct})$ where $y \in \mathcal{Y}$. This is exactly the approach we use in our experiments, with $p = 0.9$ for nucleus sampling and $k = 50$.

**Models**  We use the 7B and 13B parameter models from the LLaMA family (Touvron et al., 2023) and perform 16-bit inference for all our experiments.

**Prompting for correct and incorrect answers**  In our work, unless otherwise specified, conditioning on $(x, \mathtt{correct})$ for the $P_\mathsf{LM}$ corresponds to the standard zero-shot prompt. Similarly, conditioning on $(x, \mathtt{incorrect})$ is similar to $(x, \mathtt{correct})$ with the only difference that *"Answer:"* is replaced with *"Incorrect Answer:"* in the prompt.

**Decoding Methods**  In the multiple-choice based datasets (ARC, RACE, HHH, MMLU), we consider the following approaches:

- **Generative Ranking (G):** This baseline (Brown et al., 2020; Touvron et al., 2023) ranks every candidate $y$ by $P_\mathsf{LM}(y \mid x, \mathtt{correct})$ and picks the top candidate. This is the standard approach used in past work. Due to implementational differences, when available, we include both official scores and our version.

- **Mutual Information Ranking (MI):** This mutual-information based (Li & Jurafsky, 2016) baseline is an ensemble-based approach that reweights every candidate $y$ by $P_\mathsf{LM}(y \mid x, \mathtt{correct}) \cdot P_\mathsf{LM}(\mathtt{correct} \mid x, y)$.

- **Self-Contrastive Ranking (SC):** This approach utilizes the normalized generator $\pi_\mathsf{G}^{(1)}$ to reweight every candidate $y$ by $\pi_\mathsf{G}^{(1)}(\mathtt{correct} \mid x, y)$.

- **Discriminative Ranking (D):** This approach reweights every query-candidate pair $(x, y)$ by $\pi_\mathsf{D}^{(1)}(\mathtt{correct} \mid x, y)$.

- **Equilibrium Ranking Generator (ER-G):** Similar to **SC**, this approach utilizes the final EQUILIBRIUM-RANKING-based generator $\pi_\mathsf{G}^*$ to reweight every candidate $y$ by $\pi_\mathsf{G}^*(y \mid x, \mathtt{correct})$.

- **Equilibrium Ranking Discriminator (ER-D):** Similar to **D**, this approach utilizes the final EQUILIBRIUM-RANKING-based discriminator $\pi_\mathsf{D}^*$. This approach reweights every query-candidate pair $(x, y)$ by $\pi_\mathsf{D}^*(\mathtt{correct} \mid x, y)$.

In free-form text generation tasks (TruthfulQA, GSM8K), we additionally consider **greedy decoding**. In the mathematical reasoning task (GSM8K), we also consider **self-consistency** (Wang et al., 2022).

---

[3]As remarked at the end of the previous section, each iteration of the learning process requires a number of floating-point operations that is linear in the number $|\mathcal{Y}|$ available to the GENERATOR. In most of our settings, $|\mathcal{Y}| = 4$, making the overhead from the learning dynamics on the CONSENSUS GAME negligible compared to the cost of inference for the language model. As such, even with an unoptimized implementation of the dynamics (1,2), we observe that the computational cost associated with each iteration of the learning process takes about 40 microseconds on average.

| Domain | Model | G* | G | MI | SC | D | Equil. ranking ER-G | ER-D |
|---|---|---|---|---|---|---|---|---|
| MMLU | LLaMA-7B | – | 30.4 | 33.1 | 30.5 | **40.4** | 39.4 | 39.9 |
| | LLaMA-13B | – | 41.7 | 41.8 | 41.7 | 41.9 | 44.9 | **45.1** |
| ARC Easy | LLaMA-7B | 72.8 | 68.2 | 68.8 | 69.5 | 52.5 | **71.6** | 71.5 |
| | LLaMA-13B | 74.8 | 71.2 | 71.5 | 73.0 | 65.0 | 76.1 | **76.4** |
| ARC Challenge | LLaMA-7B | 47.6 | 47.3 | 47.4 | 56.5 | 42.7 | **58.7** | 58.3 |
| | LLaMA-13B | 52.7 | 51.9 | 52.1 | 59.3 | 48.5 | 61.1 | **61.4** |
| RACE Middle | LLaMA-7B | 61.1 | 57.7 | 57.7 | 60.4 | 51.5 | 63.2 | **63.5** |
| | LLaMA-13B | 61.6 | 60.1 | 60.2 | 64.8 | 58.3 | 67.9 | **68.6** |
| RACE High | LLaMA-7B | 46.9 | 46.4 | 46.3 | 53.1 | 46.0 | 56.3 | **56.4** |
| | LLaMA-13B | 47.2 | 47.9 | 48.4 | 58.9 | 55.1 | 62.1 | **62.8** |
| HHH | LLaMA-7B | – | 59.3 | 57.9 | 67.4 | 70.1 | **71.5** | **71.5** |
| | LLaMA-13B | – | 60.2 | 59.7 | 57.9 | **69.2** | 61.1 | 61.1 |

Table 1: Results of the different approaches across multiple tasks. We compute the accuracies on the test set of these benchmarks. EQUILIBRIUM-RANKING outperforms other approaches on most tasks. EQUILIBRIUM-RANKING performs well, even in cases where one of GENERATOR or DISCRIMINATOR is far worse than the other. **G**: Generative Ranking, **MI**: Mutual Information Ranking, **SC**: Self-Contrastive Ranking, **D**: Discriminative Ranking, **ER-G**: Equilibrium Ranking Generator, **ER-D**: Equilibrium Ranking Discriminator. * indicates the results from Touvron et al. (2023). Colors in the table entries are assigned relative to the G baseline, according to the colorbar [colorbar: -10 -5 0 +5 +10] (differences exceeding ±10 are clipped to ±10 when calculating the colors).

**MMLU** The massive multi-task language understanding benchmark (MMLU) (Hendrycks et al., 2020) is used to measure language model's multitask accuracy. It consists of questions in the multiple choice format across a wide variety of subdomains in social sciences, humanities and STEM. We evaluate our models in the zero-shot setting following the format described in Hendrycks et al. (2020); Touvron et al. (2023) and the results are presented in the first row of Table 1. For both LLaMA-7B and LLaMA-13B, the EQUILIBRIUM-RANKING-based approaches matches or outperforms all other baselines. In fact, zero-shot LLaMA-7B with ER-D (39.9) outperforms 5-shot LLaMA-7B (35.1), while zero-shot LLaMA-13B with ER-D (45.1) is competitive with 5-shot LLaMA-13B (46.9). LLaMA-7B with ER-D (39.9) even outperforms zero-shot GPT3-175B (37.7) (Hendrycks et al., 2020), while zero-shot LLaMA-13B with ER-D (45.1) outperforms 5-shot GPT3-175B (43.9) (Hendrycks et al., 2020).

**ARC** The AI2 Reasoning Challenge (ARC) (Clark et al., 2018) is an advanced question answering dataset used to study a model's knowledge and reasoning abilities based on grade school science questions. It is split in to two subcategories: easy (ARC-Easy) and challenge (ARC-Challenge). The challenge set was constructed as the set of questions that were answered incorrectly by retrieval and word co-occurence based algorithms. The results are presented in second and third rows of Table 1. On ARC-Easy, ER-D outperforms our implementation of generative ranking. We also note that LLaMA-13B with ER-D (76.4) outperform all the baseline approaches and is even competitive with the much larger PaLM-540B model (76.6) (Chowdhery et al., 2022). On ARC-Challenge, ER-D significantly outperforms all the baseline approaches. We also note that LLaMA-7B with ER-D (58.3) and LLaMA-13B with ER-D (61.4) outperforms even the much larger models: LLaMA-65B (56.0) (Touvron et al., 2023) and PaLM-540B (53.0) (Chowdhery et al., 2022) by up to 8%. Finally, we also compare against concurrent work on contrastive decoding (CD) O'Brien & Lewis (2023); Li et al. (2022). On ARC-Easy, LLaMA-13B + ER-D (76.4) is competitive with the much larger

| Domain | Model | Greedy | MI | SC | D | Equil. ranking ER-G | ER-D |
|---|---|---|---|---|---|---|---|
| TruthfulQA | LLaMA-7B | 33.41 | 34.79 ± 0.90 | **34.91** ± 0.57 | 34.17 ± 1.19 | 34.61 ± 0.99 | 34.27 ± 0.39 |
| | LLaMA-13B | 33.05 | 36.30 ± 0.37 | 34.61 ± 1.33 | 39.05 ± 1.42 | **39.83** ± 2.20 | 38.63 ± 1.76 |

Table 2: Results on TruthfulQA (Generative). Average BLEU-Acc results on the held-out set across 5 runs. LLaMA-13B with ER-G outperforms or is on par with all baselines. **MI**: Mutual Information Ranking, **SC**: Self-Contrastive Ranking, **D**: Discriminative Ranking, **ER-G**: Equilibrium Ranking Generator, **ER-D**: Equilibrium Ranking Discriminator. ± indicates 1 standard deviation computed across 5 runs. Colors are as in Table 1, relative to the Greedy baseline.

LLaMA-65B + CD ($\beta = 1.0$) (76.9). On ARC-C, we additionally note that LLaMA-13B + ER-D (61.4) outperforms LLaMA-65B + CD ($\beta = 1.0$) (59.7).

**RACE** RACE is a reading comprehension benchmark introduced in Lai et al. (2017) collected from English examinations of middle and high school students. The dataset is correspondingly split into RACE-middle and RACE-high. The dataset consists of a passage followed by questions. The passages were constructed for evaluating student's English reasoning and understanding ability. The results on this benchmark is presented in rows 4 and 5 of Table 1. On RACE-middle, like before, ER-D based models outperforms all the baselines. We note that LLaMA-13B with ER-D (68.6) even outperforms much larger models: LLaMA-65B (67.9) (Touvron et al., 2023) and PaLM-540B (68.1) (Chowdhery et al., 2022). On RACE-high, we have a similar story as with ARC-C. ER-D outperforms all baselines. LLaMA-7B with ER-D (56.4) is able to significantly outperform much larger models: LLaMA-65B (51.6) (Touvron et al., 2023) and PaLM-540B (49.1) (Chowdhery et al., 2022).

**HHH** HHH (Helpful, Honest and Harmless) (Srivastava et al., 2023; Askell et al., 2021) is a dataset of 200 multiple-choice designed to measure LM alignment with high-level quality guidelines. Here we use a different set of prompts for the GENERATOR (see Appendix B). Results are presented in the last row of Table 1. LLaMA-7B with ER-D outperforms baselines; although LLaMA-13B with ER-D with the default parameter performs worse than discriminative ranking (D) (69.2), ER-D with $\lambda_G = 0.01$ and $\lambda_D = 1.0$ achieves an accuracy of 70.6%.

**TruthfulQA** TruthfulQA (Lin et al., 2022) is a benchmark consisting of over 800 questions across a multitude of domains that were crafted to encourage humans to answer them incorrectly due to misconceptions. The dataset evaluates a model's ability to not generate false answers learnt from imitation learning on text. On this task, we consider **greedy decoding** in addition to our other ranking-based approaches. In this setting, 10 candidate sequences are sampled using nucleus and top-k sampling. These candidates are then ranked based on the approaches we described earlier. The results on the test set are presented in Table 2. Based on past work (Lin et al., 2022), we measure BLEU accuracy (BLEU-Acc). For a sequence $a$, the BLEU-Acc over reference correct candidates $b_{correct}$ and reference incorrect candidates $b_{correct}$ is computed as follows:

$$\text{BLEU-Acc}(a) \coloneqq \mathbb{I}(\text{BLEU}(a, b_{correct}) > \text{BLEU}(a, b_{incorrect})) \tag{3}$$

Where $\text{BLEU}(a, b)$ computes the BLEU score (Papineni et al., 2002) of a candidate string $a$ over a set of reference candidates $b$. With LLaMA-7B, we observe only modest improvements for ER-G and ER-D over the greedy baseline. However, with LLaMA-13B, we note increased scores for both methods. This benchmark is known to exhibit negative scaling (Lin et al., 2022) (performance drop as the model size increases). The performance difference with ER-G between LLaMA-7B and LLaMA-13B shows that EQUILIBRIUM-RANKING is in fact capable of mitigating this behavior.

**GSM8K** In our last set of experiments, we consider grade-school math (GSM8K) (Cobbe et al., 2021), a popular benchmark used to study model's mathematical reasoning ability. We use this benchmark to study whether we can combine our approach with chain-of-thought (Wei et al., 2022). As we described earlier, we generate 20 candidate reasoning paths sampled using nucleus and top-k using the 8-shot setup proposed in Wei et al. (2022). We employ self-consistency (Wang et al., 2022)

| Domain | Model | Greedy | MV | MI | SC | D | Equil. ranking | |
| | | | | | | | ER-G | ER-D |
|---|---|---|---|---|---|---|---|---|
| GSM8K | LLaMA-7B | 10.8 | 14.7 ± 0.2 | 14.6 ± 0.5 | 13.4 ± 0.3 | 15.0 ± 0.6 | 13.0 ± 0.5 | **15.1** ± 0.6 |
| | LLaMA-13B | 14.9 | 22.5 ± 0.5 | 22.5 ± 0.8 | **23.1** ± 0.5 | 22.5 ± 0.6 | 22.5 ± 0.6 | 23.0 ± 0.5 |

Table 3: Average accuracy of methods on the test set of GSM8K acroos 5 runs. In all cases, except greedy, 20 candidates were sampled. EQUILIBRIUM-RANKING-based approaches performs on par or slightly better compared to the majority vote baseline. **MV**: Majority Vote, **MI**: Mutual Information Ranking, **SC**: Self-Contrastive Ranking, **D**: Discriminative Ranking, **ER-G**: Equilibrium Ranking Generator, **ER-D**: Equilibrium Ranking Discriminator. ± indicates 1 standard deviation. Colors are as in Table 1, relative to the Greedy basline.

over the candidate sequences, where we score each reasoning path with our baselines. Finally, we aggregate the scores for each answer corresponding to the reasoning paths and pick the answer with the highest score. In Table 3, we present the results. We note that EQUILIBRIUM-RANKING-based approaches performs on par or slightly better compared to self-consistency (majority vote).

**Discussion** The application of EQUILIBRIUM-RANKING-based approaches consistently yields improved results, surpassing or at least matching the performance of all baseline approaches across various tasks. This robustness is particularly interesting, as it demonstrates that EQUILIBRIUM-RANKING is adept at handling diverse scenarios, even in situations when the initial GENERATOR or DISCRIMINATOR are not effective. As EQUILIBRIUM-RANKING is a sampling strategy, it can even be combined with deliberation methods like self-consistency (Wang et al., 2022) or tree-of-thought (Yao et al., 2023). Finally, we note that EQUILIBRIUM-RANKING demonstrates computational efficiency by eliminating the need for repetitive queries to language models.

## ACKNOWLEDGEMENTS

This work was supported by the National Science Foundation under grant IIS-2212310 and a seed grant from the MIT Schwartzman College of Computing "Artificial Intelligence for Augmentation and Productivity" program.

## 4 CONCLUSION

We have presented EQUILIBRIUM-RANKING, a training-free, game theoretic approach for generating from language models (LMs). EQUILIBRIUM-RANKING reconciles scores from generative and discriminative LM decoding procedures by formulating decoding as an imperfect-information signaling game between a GENERATOR and a DISCRIMINATOR, and leveraging computational game solving techniques to compute approximate equilibria of this game. When applied to 6 diverse question answering benchmarks: MMLU, ARC, RACE, HHH, TruthfulQA and, GSM8K, EQUILIBRIUM-RANKING offers substantial improvements over existing generative, discriminative, and mixed decoding procedures: applying EQUILIBRIUM-RANKING to LLaMA-7B sometimes outperforms much larger LLaMA-65B and PaLM-540B models. These results highlight the usefulness of game-theoretic tools in formalizing desiderata like truthfulness and stability in language modeling. Beyond the applications studied here (which focus mainly on question answer), future work might apply this toolkit to more general tasks like long-form text generation.

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

## A   DETAILS ABOUT REGRET AND REGRET DECOMPOSITION METHODS

After each repetition $t$ of the game, each player—in this case, the GENERATOR and the DISCRIMINATOR—refines their policies, in such a way that throughout the course of time, the *regrets*

$$\text{Reg}_G^{(T)} := \max_{\pi_G^*} \left\{ \sum_{t=1}^{T} u_G(\pi_G^*, \pi_D^{(t)}) - \sum_{t=1}^{T} u_G(\pi_G^{(t)}, \pi_D^{(t)}) \right\}, \tag{4}$$

$$\text{Reg}_D^{(T)} := \max_{\pi_D^*} \left\{ \sum_{t=1}^{T} u_D(\pi_G^{(t)}, \pi_D^*) - \sum_{t=1}^{T} u_D(\pi_G^{(t)}, \pi_D^{(t)}) \right\}, \tag{5}$$

cumulated by the players are guaranteed to grow sublinearly as a function of the number of rounds of learning $T$.

As mentioned in the body, a classic observation in the theory of imperfect-information sequential games is that minimization of regret (viewed as a function of their overall policy on the game tree) can be achieved by solving separate, *local*, regret minimization problems at each information set (*i.e.*, decision point) of the game. In our case, these techniques enable us to decompose the policy update of the players into separate updates for each correctness parameter $v$ (for the GENERATOR) and for each sequence $y$ (for the DISCRIMINATOR). More specifically, suppose that the GENERATOR updates their policies $\pi_G^{(t)}(\cdot \mid x, v)$ independently for each correctness parameter $v \in \{\texttt{correct}, \texttt{incorrect}\}$ they might receive, seeking to independently minimize regret

$$\text{Reg}_G^{(T)}(v) := \max_{\pi^* \in \Delta(\mathcal{Y})} \left\{ \sum_{t=1}^{T} \tilde{u}_G^{(t)}(\pi^* \mid x, v) - \tilde{u}_G^{(t)}(\pi_G^{(t)}(\cdot \mid x, v) \mid x, v) \right\}$$

with respect to the following *counterfactual utility functions*

$$\tilde{u}_G^{(t)}(\pi_G \mid x, v) := -\lambda_G \text{D}_{\text{KL}}\left( \pi_G(\cdot \mid x, v) \,\middle\|\, \pi_G^{(0)}(\cdot \mid x, v) \right) + \frac{1}{2} \sum_{y \in \mathcal{Y}} \pi_D^{(t)}(v \mid x, y) \cdot \pi_G(y \mid x, v) \tag{6}$$

for all $v$. Then, it is known that when these independent goals are met for all $v$, so is the goal of keeping regret (4) subliner, and in particular

$$\text{Reg}_G^{(T)} \leq \text{Reg}_G^{(T)}(\texttt{correct}) + \text{Reg}_G^{(T)}(\texttt{incorrect})$$

no matter the time horizon $T$. Similarly, when the DISCRIMINATOR seeks to update their policy $\pi_D^{(t)}(\cdot \mid x, y)$ for each $y \in \mathcal{Y}$ independently, so as to minimize regret

$$\text{Reg}_D^{(T)}(y) := \max_{\pi^* \in \Delta(\{\texttt{correct}, \texttt{incorrect}\})} \left\{ \sum_{t=1}^{T} \tilde{u}_D^{(t)}(\pi^* \mid x, y) - \tilde{u}_D^{(t)}(\pi_D^{(t)}(\cdot \mid x, y) \mid x, y) \right\}$$

with respect to the counterfactual utility functions

$$\tilde{u}_D^{(t)}(\pi_D \mid x, y) := -\lambda_D \text{D}_{\text{KL}}\left( \pi_D(\cdot \mid x, y) \,\middle\|\, \pi_D^{(0)}(\cdot \mid x, y) \right) + \frac{1}{2} \sum_{\substack{v \in \{\texttt{correct}, \\ \texttt{incorrect}\}}} \pi_G^{(t)}(v \mid x, y) \cdot \pi_D(v \mid x, y),$$

then their overall regret $\text{Reg}_D^{(T)}$ satisfies

$$\text{Reg}_D^{(T)} \leq \sum_{y \in \mathcal{Y}} \text{Reg}_D^{(t)}(y).$$

The counterfactual utilities $\tilde{u}_G$ and $\tilde{u}_D$ defined above are composed of a bilinear term and a strongly convex KL-regularization term. To guarantee sublinear regret with respect to such utility functions, we use the piKL algorithm Jacob et al. (2022).

A.1 GUARANTEES OF THE PIKL NO-REGRET DYNAMICS

We recall the following known properties of the piKL algorithm. An informal description of the guarantees was included in Section 2.

**Proposition 1** (Bakhtin et al. 2023). *Let $i \in \{\text{GENERATOR}, \text{DISCRIMINATOR}\}$ be any player. After any $T$ iterations of training, the regret $\text{Reg}_i^{(T)}$ cumulated by the policies $\pi_i^{(t)}$ of player $i$ produced by piKL, is upper bounded by only a logarithmic quantity in $T$. More precisely, $\text{Reg}_i^{(T)} = O\left(\min\left\{\frac{2\log T}{\lambda_i}, T\eta_i\right\} + \frac{\log|\mathcal{Y}|}{\eta_i}\right)$, where the asymptotic $O$ notation hides constants independent on the time horizon $T$, learning rate $\eta_i$ of the player, and regularization coefficient $\lambda_i$.*

**Proposition 2** (Folklore connection between regret minimization and equilibria). *The empirical frequency of play converges to the set of coarse correlated equilibria of the CONSENSUS GAME.*

**Proposition 3** (Jacob et al. 2022). *The average policy $\bar{\pi}_i^{(T)}$ of player $i \in \{\text{GENERATOR}, \text{DISCRIMINATOR}\}$ produced by piKL after $T$ iterations is guaranteed to be within a radius proportional to $\lambda_i$ centered in the initial policy $\pi_i^{(1)}$. More precisely, $\text{D}_{\text{KL}}(\bar{\pi}_i^{(T)}, \pi_i^{(1)}) \leq \frac{1}{\lambda_i}(1 + o(1))$, where the asymptotic notation $o(1)$ denotes a quantity decreasing, as a function of the time horizon $T$, at a rate of $\log T/(\lambda_i T)$.*

# B HHH PROMPTS

In the HHH experiments, $(x, \text{correct})$ corresponds to the prompt:
```
"You are a helpful, honest and harmless assistant.
Human: {x}
Assistant:"
```

And $(x, \text{incorrect})$ corresponds to the prompt:
```
"You are a unhelpful, dishonest and harmful assistant.
Human: {x}
Assistant:"
```

# C OTHER RELATED WORK

Many decoding strategies have been proposed for language models, such as top-k sampling (Fan et al., 2018a), nucleus sampling (Holtzman et al., 2020), and typical sampling (Meister et al., 2023). These methods primarily focus on producing diverse, high-quality text from a language model. However, they decode from the LM without any emphasis on the correctness of the generated sequences. As we show in Section 3, EQUILIBRIUM-RANKING is naturally complementary and be combined with any of these sampling strategies.

Re-ranking is a common approach for selecting the correct answer from a set of candidates sampled from LM. Cobbe et al. (2021) train a verifier to re-ranked the sampled outputs. Shen et al. (2021) jointly train a ranking model with the generation model to improve the model accuracy. Thoppilan et al. (2022) collect additional human annotations to train the ranking model for response filtering. As we discuss in Section 2, our work focuses on leveraging an existing LM and using them in a training-free manner as a discriminator. However, we note that we do not make any specific assumption on the specific form of a the GENERATOR or DISCRIMINATOR. As such, EQUILIBRIUM-RANKING can be combined with these approaches.

As previously mentioned, EQUILIBRIUM-RANKING differs from recent deliberation methods, as highlighted in various recent work (Wei et al., 2022; Madaan et al., 2023; Shinn et al., 2023; Yao et al., 2023; Dohan et al., 2022). In Section 3, we demonstrate how EQUILIBRIUM-RANKING can be integrated with these approaches. In another line of work, Du et al. (2023) and Chen et al. (2023a) employ multiple instances of language models suggest and "debate" individual responses and reasoning processes across multiple iterations, ultimately converging on a shared final answer. In contrast, EQUILIBRIUM-RANKING can be viewed as a variant of this multi-agent debate, wherein the "debate" occurs within the regret-minimization framework rather than in the context of language models.

# D    ADDITIONAL DISCUSSION

## D.1    RELATIVE CONTRIBUTIONS OF NORMALIZATION AND EQUILIBRIUM SEARCH

To tease apart these two components of EQUILIBRIUM-RANKING, we run an additional ablation experiment that re-ranks using a pointwise-mutual-information-style Li & Jurafsky (2016) product using $\pi_G^{(1)}$ and $\pi_D^{(1)}$ and rather than unnormalized generator (G) and discriminator (D) probabilities. This new ablation (labled MI* in the table below) does improves performance over baseline approaches but consistently underperforms the full EQUILIBRIUM-RANKING method.

| Domain | Model | MI* | ER-G | ER-D |
|---|---|---|---|---|
| MMLU | LLaMA-7B | 35.8 | 39.4 | **39.9** |
| | LLaMA-13B | 43.1 | 44.9 | **45.1** |
| ARC-Easy | LLaMA-7B | 71.04 | **71.6** | 71.5 |
| | LLaMA-13B | 76.1 | 76.1 | **76.4** |
| ARC-Challenge | LLaMA-7B | 58.5 | **58.7** | 58.3 |
| | LLaMA-13B | **61.4** | 61.1 | **61.4** |
| RACE-Middle | LLaMA-7B | 62.8 | 63.2 | **63.5** |
| | LLaMA-13B | 67.5 | 67.9 | **68.6** |
| RACE-High | LLaMA-7B | 55.9 | 56.3 | **56.4** |
| | LLaMA-13B | 62.2 | 62.1 | **62.8** |
| HHH | LLaMA-7B | 69.7 | **71.5** | **71.5** |
| | LLaMA-13B | **61.1** | **61.1** | **61.1** |

Table 4: Results of different approaches across multiple tasks. In particular, we consider an additional baseline MI* that improves performance over baseline approaches but consistently underperforms the full EQUILIBRIUM-RANKING. Colors are as in Table 1, relative to the MI* ablation.

## D.2    ANALYSIS OF COHERENCE

In order to look at how severe the problem of coherence between discriminative and generative methods are, we perform an analysis looking at how often the answers chosen by Generative Ranking (G) and Discriminative Ranking (D) disagree in each task. Table 5 shows that they do in fact disagree with each other a significant amount. Furthermore, we also observe that in cases where the disagreement % is larger, EQUILIBRIUM-RANKING offers the most benefit relative to G (The pearson correlation between "Disagreement %" and "% Improvement" is 0.64).

| Domain | Model | Disagreement % (G & D) | G | ER-D | % Improvement |
|---|---|---:|---|---|---:|
| MMLU | LLaMA-7B | 69 | 30.4 | 39.9 | **31.3** |
| | LLaMA-13B | 60.6 | 41.7 | 45.1 | **8.1** |
| ARC-Easy | LLaMA-7B | 56.1 | 68.2 | 71.5 | **4.8** |
| | LLaMA-13B | 46.1 | 71.2 | 76.4 | **7.3** |
| ARC-Challenge | LLaMA-7B | 65.9 | 47.3 | 58.3 | **23.2** |
| | LLaMA-13B | 59.1 | 51.9 | 61.4 | **18.3** |
| RACE-Middle | LLaMA-7B | 55.8 | 57.7 | 63.5 | **10.0** |
| | LLaMA-13B | 53.2 | 60.1 | 68.6 | **14.1** |
| RACE-High | LLaMA-7B | 62 | 46.4 | 56.4 | **21.5** |
| | LLaMA-13B | 58.8 | 47.9 | 62.8 | **31.1** |
| HHH | LLaMA-7B | 46.1 | 59.3 | 71.5 | **20.5** |
| | LLaMA-13B | 38 | 60.2 | 61.1 | **1.5** |

Table 5: Comparison of how often the answers chosen by Generative Ranking (G) and Discriminative Ranking (D) disagree in each task and the relative percentage improvement of EQUILIBRIUM-RANKING-DISCRIMINATOR (ER-D) over G.

