# OpenReview forum: "The Consensus Game: Language Model Generation via Equilibrium Search"
_ICLR.cc/2024/Conference — ICLR 2024 spotlight_

### Official Review · Reviewer_ji8z · 2023-10-26

**Soundness:** 3 good
**Presentation:** 3 good
**Contribution:** 3 good
**Rating:** 6
**Confidence:** 3

**Summary:**

The paper introduces a new decoding algorithm called EQUILIBRIUM-RANKING, which addresses the challenge of reconciling different scoring procedures in language models (LMs) for question-answering and text-generation tasks. EQUILIBRIUM-RANKING is a game-theoretic approach, where a generator aims to communicate a correctness parameter to a discriminator through natural language sentences. By formulating language model decoding as an imperfect information sequential signaling game, the authors develop computational procedures to find approximate equilibria of the game. The experiment uses a set of QA tasks to evaluate EQUILIBRIUM-RANKING, including MMLU, ARC, RACE, HHH, and TruthfulQA, and a math benchmark GSM8K. Across these tasks, EQUILIBRIUM-RANKING consistently improves performance compared to existing LM decoding procedures.

**Strengths:**

1. The paper demonstrates originality by introducing a novel approach to decoding language models through the consensus game framework. The idea of casting language model decoding as a game-theoretic problem and seeking approximate equilibria is innovative and provides a fresh perspective on addressing the challenge of reconciling different scoring procedures.

2. The authors develop robust computational techniques for finding approximate equilibria in the consensus game, ensuring reliable and practical implementation of the EQUILIBRIUM-RANKING algorithm. The paper also demonstrates the quality of the research through its extensive evaluation across a diverse range of tasks, showcasing consistent performance improvements over existing decoding procedures.

3. The authors provide a concise and coherent description of the EQUILIBRIUM-RANKING algorithm, making it accessible to readers. The clarity of the writing facilitates the comprehension and replication of the proposed approach. The paper also presents the results and performance improvements in a straightforward manner, enabling readers to grasp the significance of the findings.

**Weaknesses:**

1. To improve the clarity of the proposed method, it would be advantageous to include a running example that demonstrates the step-by-step process.

2. To enhance the quality of the paper, it is recommended to address minor typos and errors. For instance, in line 6 of the abstract, "a new, a training-free" should be corrected to "a new, training-free." Additionally, in section 2, line 6, "we may them" appears to be a typographical error and should be revised for clarity.

**Questions:**

How is the payoff matrix defined for the consensus game?

---

> ### Author Response · Authors · 2023-11-17
> **Response to Reviewer ji8z**
>
> Thank you for your review!
>
> **Clarity of the proposed method**
>
> Thank you for the suggestion. To improve clarity, we will include a pseudo-code with a running example in our revision.
>
> **Typos**
>
> Thank you for catching those typos. We will fix them in our revision.
>
> **How is the payoff matrix defined for the consensus game?**
>
> Signalling games are sequential in nature, as such, they are typically represented in tree-form as we show in Figure 1 (i.e., D and G both receive a reward of 1 each if D picks the correctness parameter that D was trying to communicate. In all other cases, they receive a reward of 0 each). More specifically, D and G receive a reward of (1, 1) each if and only if G picks the same correctness parameter (correct or incorrect) that D received. In all other cases, they receive (0, 0). So, for example, if D receives the signal "correct" and produces a natural language string "u". And if G, conditioned on "u", produces "correct", they both receive (1, 1). If G instead picks "incorrect", they both receive (0,0).
>
> We hope we have addressed your questions and concerns. Do let us know if you have any other questions!

---

### Official Review · Reviewer_XHXa · 2023-10-30

**Soundness:** 3 good
**Presentation:** 3 good
**Contribution:** 3 good
**Rating:** 8
**Confidence:** 4

**Summary:**

This work proposes a new answer reranking method for LLMs. The main objective is to reconcile the inconsistency between the model's response in the generator setting and its response in the discriminator setting. Specifically, this work designs a regularized imperfect-information sequential signaling game involving the generator and discriminator and finds the Nash equilibrium between them. To find the equilibrium, this work adopts the piKL algorithm proposed by Jacob et al., 2022. The overall method is a training-free method and can consistently outperform several baselines on a variety of QA tasks (also including open-ended and math reasoning tasks).

**Strengths:**

1. The inconsistent behavior of LLMs across different prompts is an annoying phenomenon. This paper proposes a principled method to reconcile such consistency via game theory.
2. The proposed method is evaluated on various tasks and consistently improves over popular baselines.
3. The proposed method is training-free, so it's applicable to even very large-scale models.

**Weaknesses:**

1. If we compare the proposed ER-based reranking method and previous baselines, there are actually two major differences. One is the process of finding the Nash equilibrium under the regularization part, the other is the combination of two normalized probabilities in Section 2.2. In the current evaluation, it is clear that the combination of these two can bring consistent improvement, but it's unclear what is the effect of each individual component. It would be helpful to have a baseline just combining the two normalized probabilities (e.g., via the MI approach).
2. This paper can benefit from showing some intrinsic analysis of the inconsistency of the generator and the discriminator. Right now, while the evaluation shows the effectiveness of the proposed approach, it is unclear how severe the problem is, and in how many cases finding the equilibrium can help.
3. I understand the piKL algorithm is from Jacobs et al., 2022, but I would still suggest the authors provide a slightly more detailed description in the appendix. The lack of related details makes it very hard to understand how important the hyper-parameters are, and if the method is sensitive to these values.

**Questions:**

1. I'm trying to understand equations (1) and (2) on page 5, but I can't really understand how the weights terms (i.e., $\lambda_{G}$  and $\eta_Gt$ part) work. Should there be an additional $1/t$ before the $Q$? This will also make it more similar to the algorithm in Jacob et al., 2022. Otherwise, can you explain why there is a difference?
2. Since this is a training-free method, I'm wondering if the authors have tried on even larger models.
3. This can entirely be future work for a separate paper, but I just wonder if the authors have tried to fine-tune the models (e.g., the smaller 7B one) using the same objective.

Typo: on page 7, Race-high -> RACE-high

---

> ### Author Response · Authors · 2023-11-17
> **Response to Reviewer XHXa -- Part 1 / 2**
>
> Thank you for your review!
>
> **What are the relative contributions of normalization and equilibrium search?**
>
> This is a great question! To tease apart these two components of EQUILIBRIUM-RANKING, we have run a new ablation experiment that reranks using a pointwise-mutual-information-style product using $\pi_G^{(1)}$ and $\pi_D^{(1)}$ rather than unnormalized generator and discriminator probabilities. This new ablation (labled MI* in the table below) does improves performance over  baseline approaches but consistently underperforms the full EQUILIBRIUM-RANKING method. We will include these results in the revised version of the paper that we will upload in the next few days.
>
>
> | Domain        | Model | ER-G     | ER-D     | MI\*     |
> | ------------- | ----- | -------- | -------- | -------- |
> | MMLU          | 7B    | 39.4     | **39.9** | 35.8     |
> |               | 13B   | 44.9     | **45.1** | 43.1     |
> | ARC-Easy      | 7B    | **71.6** | 71.5     | 71.04    |
> |               | 13B   | 76.1     | **76.4** | 76.1     |
> | ARC-Challenge | 7B    | **58.7** | 58.3     | 58.5     |
> |               | 13B   | 61.1     | **61.4** | **61.4** |
> | RACE-Middle   | 7B    | 63.2     | **63.5** | 62.8     |
> |               | 13B   | 67.9     | **68.6** | 67.5     |
> | RACE-High     | 7B    | 56.3     | **56.4** | 55.9     |
> |               | 13B   | 62.1     | **62.8** | 62.2     |
> | HHH           | 7B    | 71.5     | **71.5** | 69.7     |
> |               | 13B   | 61.1     | 61.1     | 61.1     |
>
> **Intrinsic Analysis of Inconsistency**
>
> Thank you for the suggestion! In order to look at how severe the problem is, we performed an analysis looking at how often the argmax answers from G and D disagree in each task. The table below shows that they do in fact disagree with each other a significant amount. Furthermore, we also observe that in cases where the disagreement % is larger, EQUILIBRIUM-RANKING offers the most benefit relative to G (The pearson correlation between "Disagreement %" and "% Improvement" is 0.64). We will include these results in the revised version of the paper that we will upload in the next few days.
>
>
> | Task           | Model | Disagreement % (G & D)   | G    | ER-D | % Improvement   |
> |----------------|-------|--------------------------|------|------|-----------------|
> | MMLU           | 7B    | 69                       | 30.4 | 39.9 | 31.3            |
> |                | 13B   | 60.6                     | 41.7 | 45.1 | 8.1             |
> | ARC-Easy       | 7B    | 56.1                     | 68.2 | 71.5 | 4.8             |
> |                | 13B   | 46.1                     | 71.2 | 76.4 | 7.3             |
> | ARC-Challenge  | 7B    | 65.9                     | 47.3 | 58.3 | 23.2            |
> |                | 13B   | 59.1                     | 51.9 | 61.4 | 18.3            |
> | RACE-Middle    | 7B    | 55.8                     | 57.7 | 63.5 | 10.0            |
> |                | 13B   | 53.2                     | 60.1 | 68.6 | 14.1            |
> | RACE-High      | 7B    | 62                       | 46.4 | 56.4 | 21.5            |
> |                | 13B   | 58.8                     | 47.9 | 62.8 | 31.1            |
> | HHH            | 7B    | 46.1                     | 59.3 | 71.5 | 20.5            |
> |                | 13B   | 38                       | 60.2 | 61.1 | 1.5             |
>
>
> **More detailed discussion of piKL in the appendix?**
>
> Thank you for your suggestion, we will do that!
> More context behind the choice and properties of piKL can be found in our response to Reviewer Jsc5.

---

> ### Author Response · Authors · 2023-11-17
> **Response to Reviewer XHXa -- Part 2 / 2**
>
> **Question 1 (Equations (1) and (2))**
>
> The difference between the way our algorithm is written, and the way piKL-Hedge (Algorithm 1 of Jacob et al.) is written, is just cosmetic and amounts to a change of variable. The policies the two algorithms produce are equivalent.
>
> In the original piKL paper, Algorithm 1 (piKL-Hedge) keeps track of cumulative values in the variable named $\textsf{CV}^{(t)}$ on line 11. Then, the update is
>
> $$ \pi^{(t+1)}(a) \propto \exp \{ \frac{\eta \textsf{CV}^{(t)}(a) + t \lambda \eta \log \tau(a)} {1 + t\lambda\eta} \}$$
>
> (where for us the anchor policy $\tau$ is $\pi^{(1)}$, the initial policy of the language model.) In our paper, we use the same formula above, but divide the numerator and denominator of the fraction inside the exponentiation by the same quantity $\eta t$. This results in an update of the form
> $$ \pi^{(t+1)}(a) \propto \exp \{ \frac {(\frac{1}{t}\textsf{CV}^{(t)}(a)) + \lambda \log \tau(a) } {1/(\eta t) + \lambda } \}$$
>
> In fact, in our paper we keep track of the average values, which we call $Q$ (indeed, observe in the display equation at the very bottom of page 4, the cumulated values $\sum\frac{1}{2}\pi(\cdots)$ are divided by $t$ on the left of the sum). So, in summary, our updates are equivalent to those of piKL and can be written as
> $$ \pi^{(t+1)}(a) \propto \exp \{ \frac {Q^{(t)}(a) + \lambda \log \tau(a) } {1/(\eta t) + \lambda } \}$$
> which is what we have in the paper.
>
> The reason why we performed this change is that the numerical range of the $Q$ variables we keep track of remains small, whereas the cumulated utilities $\textsf{CV}$ can grow linearly in absolute value.
>
> **Question 2 (What about larger models?)**
>
> Thank you for the suggestion. Unfortunately, due to the size of the datasets and the number of tasks involved, we have not been able to evaluate our approach on models larger than LLaMA-13B. We leave this to future work!
>
> **Question 3 (Could ER be used for fine-tuning?)**
>
> We think the idea of fine-tuning the models using this objective is a great idea, but this is not something we have been able to try yet.
>
> **Typo**
>
> Thanks for the catch. We will fix this in our revision.

---

> > ### Comment · Reviewer_XHXa · 2023-11-21
> >
> > Thank the authors for the additional experiments and the clarifications. The response resolved many weaknesses in my original review. I believe this paper will be substantially clearer and stronger with these additional results in the final version. That said, the improvement is slightly underwhelming (<1% in 8/12 experiments) compared to the new baseline. In such cases, it would be great if the authors can provide some insight on when and how the ER-D/ER-G methods can bring substantial improvement. For that reason, I feel this paper is more suitable for a score of 7 instead of 8, However, 7 is not an option available, so I keep my score unchanged.

---

> > > ### Author Response · Authors · 2023-11-22
> > > **Follow-up Response to Reviewer XHXa**
> > >
> > > Thank you for your response! We really appreciate that you are engaging with us in constructive discussion that is allowing us to strengthen this paper further.
> > >
> > > **Magnitude of improvement**
> > >
> > > (Thanks for the intended increase in score!) We hope that the following additional considerations provide additional clarification about the MI* experiments:
> > >
> > > 1. The formulas used to compute initial $\pi_G^{(1)}$ and $\pi_D^{(1)}$ are themselves an independent contribution of this work and stem from the specific game-theoretic formulation we propose (based on transmitting an abstract correctness parameter). So, it would be more accurate to characterize MI* more as an *ablation* than an additional *baseline*.
> > >
> > > 2. Our data shows that performing equilibrium search on top of the anchors is always better than not doing so. In other words, allowing the learning agents to explore within some reasonable radius from the initial policies is beneficial across the variety of tasks we experiment on, and in some cases it is in fact significantly better. Arguably, this points pretty strongly towards the benefit of equilibrium refinement as a default component that can be enabled in the loop.
> > >
> > > 3. The results for ER-D and ER-G in the paper are for the default values of $\lambda_G = \lambda_D = 0.1$ we use across all domains. We kept this consistent to demonstrate that our approach does not require extensive hyperparameter search and consistently beats (sometimes significantly) the baselines. In subsequent experiments, we have found that by tuning the $\lambda$ parameters, the gains could be even larger. For example, we know that in HHH, setting $\lambda_G = 0.01$ and $\lambda_L = 1.0$ leads to a substantial improvement in performance for ER-D (61.1 $\to$ 70.6).
> > >
> > > We will add this discussion to our final revision.
> > >
> > > **Insight on when and how the ER-D/ER-G methods improves over ablations and baselines**
> > >
> > > Consider the table below that computes the conditional probability that $\pi_G^{(1)}$ and $\pi_D^{(1)}$ pick the wrong answer conditional on them agreeing.
> > >
> > > | Task            | Model | ER-G | ER-D | MI*  | P(answer is incorrect \| $\pi_G^{(1)}$ and $\pi_D^{(1)}$ agree on the same answer) |
> > > |-----------------|-------|------|------|------|---------------------------|
> > > | MMLU            | 7B    | 39.4 | 39.9 | 35.8 | 0.528              |
> > > |                 | 13B   | 44.9 | 45.1 | 43.1 | 0.426              |
> > > | ARC-Easy        | 7B    | 71.6 | 71.5 | 71.04| 0.161              |
> > > |                 | 13B   | 76.1 | 76.4 | 76.1 | 0.096             |
> > > | ARC-Challenge   | 7B    | 58.7 | 58.3 | 58.5 | 0.303              |
> > > |                 | 13B   | 61.1 | 61.4 | 61.4 | 0.253               |
> > > | RACE-Middle     | 7B    | 63.2 | 63.5 | 62.8 | 0.252                |
> > > |                 | 13B   | 67.9 | 68.6 | 67.5 | 0.178              |
> > > | RACE-High       | 7B    | 56.3 | 56.4 | 55.9 | 0.309              |
> > > |                 | 13B   | 62.1 | 62.8 | 62.2 | 0.231              |
> > > | HHH             | 7B    | 71.5 | 71.5 | 69.7 | 0.182              |
> > > |                 | 13B   | 61.1 | 61.1 | 61.1 | 0.279              |
> > >
> > >
> > > Generally speaking, there seems to be a weak correlation such that the highest performance increase for ER relative to MI* corresponds to those cases in which G and D were agreeing on the **wrong** answer. In MMLU - 7B, G and D have the highest conditional probability of 0.528 and this is also where ER-D has the highest relative improvement over MI* by 11.45%. Similarly, in MMLU - 13B, we see a relative improvement of 4.6% when this conditional probability is 0.42.
> > >
> > > So, we believe that a reasonable predictor of substantial improvement over MI* for EQUILIBRIUM-RANKING could be a measure of how often the unrefined $\pi_G^{(1)}$ and $\pi_D^{(1)}$ agree on the wrong answer.
> > >
> > > We hope we have addressed your questions!

---

### Official Review · Reviewer_Jsc5 · 2023-11-02

**Soundness:** 3 good
**Presentation:** 3 good
**Contribution:** 2 fair
**Rating:** 6
**Confidence:** 2

**Summary:**

In this paper, the authors present an interesting framework centered around a consensus game involving generators and discriminators within language models. They delve into the concept of equilibrium within this game and leverage the equilibrium states of the generator and discriminator to enhance output generation in language models. The efficacy of this approach is validated through rigorous experimentation across various question-answering (QA) tasks.

**Strengths:**

1. The game-theoretic formulation introduced for the interaction between generators and discriminators in language models is novel and interesting.
2. The experiments is comprehensive, covering an array of QA tasks, and the results presented are compelling.
3. The paper is generally well-written and easy to follow.

**Weaknesses:**

1. The paper could benefit from a more detailed explanation on the methodological. The process by which Equations (1) and (2) facilitate the attainment of the game's equilibrium is not sufficiently illuminated.
2. The reliance on the piKL algorithm for the learning dynamics of the consensus game may limit the methodological novelty of the paper.
3. The pursuit of a no-regret dynamic steers the system towards a coarse correlated equilibrium. The paper could be enriched by an exploration into the feasibility and potential advantages of converging to stronger equilibrium constructs, such as pure Nash or mixed Nash equilibria.
4. A notable performance drop is observed on the HHH dataset when transitioning from the LLaMA-7B to LLaMA-13B backbone model. An elaboration on this counterintuitive outcome would be beneficial.

**Questions:**

See the weakness part.

---

> ### Author Response · Authors · 2023-11-17
> **Response to Reviewer Jsc5**
>
> Thanks for your review!
>
> **Methodological notes behind the choice of Equations (1) and (2)**
>
> We use Equations (1) and (2) to iteratively refine the policies of the players (generator and discriminator). Such policy-refinement procedures typically go under the name of "learning dynamics" in the theory of learning in games. Methogologically, the learning dynamics we use in our paper were designed to achieve the following desiderata:
> 1. The update rules in Equation (1) and (2) are very easy to implement in practice, in that they only require elementary operations such as exponentiation and normalization. This enables us to efficiently implement the refinement operations in a framework such as NumPy or PyTorch in only a couple of lines of code.
> 2. The update rules guarantee strong _no-regret_ properties. Regret is the standard optimality metric for online learning, and intuitively tracks the discrepancy between realized utility and the best utility that could have been realized in hindsight.
> 3. The update rules ensure that the refined policies do not get arbitrarily far from the initial policies that are produced by the language model. As explained in the paper, this is important to guarantee _reasonableness_.
> 4. By virtue of their no-regret properties, the update rules retain strong connections with game-theoretic notions of optimality.
>
> From a technical point of view, Equations (1) and (2) are a form of mirror descent, a generalization of gradient descent that accomodates more specialized geometries, such as the natural geometry of conditional probability distributions, such as in the case of this paper.
>
> **On the reliance on piKL**
>
> We disagree that reliance on piKL limits the methodological novelty of the paper, for two reasons:
> 1. The use of game-theoretic learning dynamics for language generation is novel. For historical context, prior uses of piKL were in the direction of providing interpolations between imitation learning (aka. behavioral cloning) and pure self-play learning in the context of multi-agent reinforcement learning. However, we realized that the strong mathematical properties of piKL dynamics could be used to help guide production of language, which led to this paper. (See also above for considerations that help explain the speicifc choice of using piKL dynamics.)
> 2. From a technical point of view, piKL dynamics were defined in the context of normal-form games (that is, simultaneous-action nonsequential games, such as rock-paper-scissors). Instead, we instantiate piKL as part of a sequential potential (coordination) game.
>
> **Why search for coarse correlated equilibria?**
>
> The existing analysis of piKL-hedge provides a guarantee of convergence to a coarse correlated equilibrium. However, as we hinted at the very end of Section 2.2, we  believe that in the case at hand most no-regret dynamics (including our own) will actually converge to Nash equilibrium. This is because the consensus game is a *potential game*, and in potential games dynamics based on smooth regularization are known to lead to Nash equilibria. Unfortunately, piKL is not, strictly speaking, an instance of such a dynamics, due to the fact that entropic regularization does not guarantee Lipschitz-continuous gradients. (Very small perturbations of piKL dynamics could fix the issue, but introducing a negligible amount of $\epsilon$-exploration). In practice, we observe excellent convergence properties, and leave the question of ironing out these technical subtleties for future work.
>
> We will add this discussion in the next version of the paper that we will upload in the next few days.
>
> **Drop in performance on the HHH dataset**
>
> We first note that on this benchmark, other hyperparameter settings of EQUILIBRIUM-RANKING for LLaMA-13B results in the performance that is close to LLaMA-7B. The drop in performance for the larger model could be explained by the fact that HHH dataset tests the alignment of instruction-tuned chatbots. The LLaMA models that we use are not instruction-tuned which makes them poor at following prompts of the form listed in Appendix B. Furthermore, inverse-scaling (where, performance of larger models degrade) has been observed in similar tasks such as TruthfulQA [1].
>
> [1] Lin, Stephanie, Jacob Hilton, and Owain Evans. "TruthfulQA: Measuring How Models Mimic Human Falsehoods." Proceedings of the 60th Annual Meeting of the Association for Computational Linguistics (Volume 1: Long Papers). 2022.

---

> > ### Comment · Reviewer_Jsc5 · 2023-11-21
> >
> > I thank the authors for the response. I will keep my score unchanged.

---

> ### Author Response · Authors · 2023-11-21
> **Response to Reviewer Jsc5**
>
> Thank you for your response! We have attempted to address all the weaknesses above. Are there any aspects you don't think have been addressed, and is there anything else we could do to address these concerns in our revision?

---

### Official Review · Reviewer_DG2h · 2023-11-04

**Soundness:** 4 excellent
**Presentation:** 3 good
**Contribution:** 4 excellent
**Rating:** 10
**Confidence:** 4

**Summary:**

The paper introduces a new decoding strategy for LM's based on a game theoretic formulation. It is directed at tasks requiring objective, factual answers for which a clear notion of a generative version and a discriminative version can be applied. In such settings, the paper describes an adaptation of recent regularized equilibrium approaches that operate directly on the outputs of samples from the Generator/discriminator to produce final answers that do quite well in experiments over a range of challenging fact-seeking or reasoning-seeking benchmarks.

**Strengths:**

Originality: There have a lot of recent investigations into improved decoding strategies; this paper's approach is quite original in taking a game-theoretic formulation to (essentially) the problem of factuality from language models. This actually makes a lot of sense; human language as pointed out in one of the references is a strategic game between agents and thus we should be modeling it as such and not a pure optimization problem. This particular insight is not new, but this idea stands out to me because of its practicality. It should be noted that the idea could be said to be inspired by recent work on diplomacy and the core algorithm is an adaptation of the same. However, it takes a good deal of originality to make the connection to discriminator/generator as a game and present the particular formulation in the paper.

Quality: The approach has the strong benefit of simplicity in implementation, it requires no re-training of the LM itself. It does seem to require some degree of non-trivial post-processing in the form of running the iteration procedure of eq 1/2. The authors dont address this but I would expect that it would not affect overall latency of LM inference by much and also it scales zero with LM size (iiuc). The experimental evaluation is quite exhaustive and satisfactory, there are a range of datset types explored. the equilibrium methods don't dominate on every single dataset but they generally do quite close to the SoTA and in some instances far exceed.

Clarity: The motivational sections, the presentation of the key insight and the experimental section are mostly quite clear and the paper was a pleasure to read. I appreciate the use of shading to visualize the relative improvement between the best and other methods in Table 1/2/3.

Significance: I think this could be an very impactful paper. The results are generally very good, it is easy to implement and does not require extensive tuning it seems to get to work, and it seems from my understanding to be scalable to very large LM sizes.

**Weaknesses:**

Clarity: The paper is actually written fairly well, but there could be some improvements:
 a. sec 2.2 was far too short. readers without any Game theory background might not even know what regret means. You dont need to fully flesh out all the derivation and algorithm, but a better intuition I think should be built, which might require 2-3 paragaphs more. I think this can be taken from the experimental section, where although there is a lot of useful material, at a pinch some of it can be moved to appendix to make room. Further a couple of (minor) sources of confusion on my first read:

 a.  'reasonableness' is a bit of a misleading word for the concept you are trying to capture. Maybe 'alignment' ? bu that has other implications in the LM space.
 b. eq 1/2: i think it's helpful to clarify that this update does not result in a backprop to the params of the LM. The reason is that in the current RLHF literature that is generally what happens so it is easy to get confused.

c. Bit confusing to put ER-G and ER-D in the section on "Baselines".


Quality: I am not at all sure about the characterization of SC as significantly related to Contrastive Decoding. The idea of using a weaker LM as the contrastive seems quite significant to me as it enables the strong LM to "avoid" the weaker ones mistakes, which for reasoning problems often are systematic reasoning errors. So it is unfortunate that CD was not compared against (Though I guess the paper with strong results for CD on reasoning problems was quite recent?)

Significance:  The biggest challenge to the potential significance of this work is that it doesnt seem readily applicable for all use cases of LMs. You need a sort of fact-checking problem, where a set of candidate answers are present. This means it can't be used out of the box as a general decoder, which would be sort of a holy grail right now. Nevertheless, I think the set of applications it is relevant for is hugely important and it is likely to find application given its ease of use.

**Questions:**

1. Regarding the remark in the last para of sec 2.2, what is the difference between the convergence result of Anagostides (2022) and the one claimed earlier in the prev para?

2. I am wondering about a subtle form of bias that may creep in with use of this method. We "train" the model with both correct and incorrect answers at the root node, which is good. but then over time we are likely to model select on some dev set using only accuracy as our metric [since generally that's what we care about], which means we only care about the 'correct' branches. Is this a possible problem?

3. Both ER-G and ER-D are quite competitive and there is no clear winner between the 2. Is there some simple combination strategy you recommend? ensemble or averaging?

---

> ### Author Response · Authors · 2023-11-17
> **Response to Reviewer DG2h**
>
> Thank you for your review!
>
> **Clarity**
>
> 1. Thank you for your suggestion, we will improve section 2.2 to include a more detailed introduction in our revision.
> a. "Reasonableness": We acknowledge this concern and are open to suggestions! As you pointed out, "alignment" does have usage in this space that could make it confusing.
> b. We agree and will note this point in our revision.
> c. You are right. This subsection should be called "decoding methods".
>
> **What is the relationship between our CD experiments and [1]?**
>
> We agree our CD baseline is sufficiently different from other work that we should clarify in our revision. We were unable to compare against CD for reasoning [1] as it was concurrent with our work. We do note that we *can* provide a direct comparison to published CD numbers on ARC-E and ARC-C. On ARC-E, LLaMA-13B + ER-D (76.4) is competitive with the (**much larger**) LLaMA-65B + CD ($\beta=1.0$) (76.9). On ARC-C, LLaMA-13B + ER-D (61.4) outperforms LLaMA-65B + CD ($\beta=1.0$) (59.7).
>
> **How does this generalize to other language modeling problems?**
>
> Our approach does require using LMs as "sequence-level" discriminators that evaluate complete generations. That said, we are still able to apply it to standard generation tasks by formulating them as generate-and-rerank problems (like we do in GSM8K and TruthfulQA) - that is, we first sample a set of candidates from the language model using top-p and top-k sampling then use EQUILIBRIUM-RANKING on top. It can thus already be applied to open-ended generation problems. We think that recent work in partial reward models [2, 3] is a promising direction for future work that could apply it directly to generation rather than discriminative reranking.
>
> **Convergence result of Anagostides et al.**
>
> A celebrated result in the theory of learning in games is that the average play of no-regret dynamics converges to the set of coarse correlated equilibria of the game. However, since our game has additional structure (in particular, it is a potential game), most learning dynamics in fact converge to Nash equilibria, which are always a subset of coarse correlated equilibria. Anagnostides et al. provide a general framework for analyzing this phenomenon for a general class of dynamics based on the online mirror descent framework. Unfortunately, while piKL is in the spirit of the algorithms studied by Anagnostides at al., we remark that due to technical conditions (specifically, a lack of Lipschitz continuity of the gradient of the entropy regularizer close to the boundary of the policy space), piKL is not, strictly speaking, an instance of the dynamics studied by Anagnostides et al. In practice, this subtlety seems to be simply a technical detail, and we indeed observe excellent convergence properties. We leave the question of ironing out these technicalities for future work.
>
> **Bias in approach**
>
> In the gameplay itself, the "correctness parameter" is always uniformly sampled from between "correct" and "incorrect". However, we haven't experimented with fitting this prior on a dev set, which as you pointed out could result in bias of the form you described.
>
> **Combining ER-G and ER-D**
>
> We do not have strong intuition here but we think that additive or multiplicative ensembling are both great options for combining these two approaches!
>
> [1] O'Brien, Sean, and Mike Lewis. "Contrastive decoding improves reasoning in large language models." arXiv preprint arXiv:2309.09117 (2023).
>
> [2] Wu, Zeqiu, et al. "Fine-Grained Human Feedback Gives Better Rewards for Language Model Training." arXiv preprint arXiv:2306.01693 (2023).
>
> [3] Yang, Kevin, and Dan Klein. "FUDGE: Controlled Text Generation With Future Discriminators." Proceedings of the 2021 Conference of the North American Chapter of the Association for Computational Linguistics: Human Language Technologies. 2021.

---

> > ### Comment · Reviewer_DG2h · 2023-11-21
> > **thanks**
> >
> > Thank you for your response ! I have no further q's.

---

### Author Response · Authors · 2023-11-17
**Note from Authors**

Thank you all for your thorough reviews!

We're gratified that you found the paper "very impactful" (DG2h), "principled" (XHXa), "innovative" (ji8z), and "novel and interesting" (Jsc5).
We believe that your suggestions have helped improve the manuscript. In the next few days, we will submit a revised version that incorporates your comments.

In the meantime, all reviewers may be interested in a set of new baselines and comparison we have run at the suggestion of DG2h and XHXa, which (1) show that ER outperforms concurrent results from Contrastive Decoding methods in larger models, and (2) investigate a ablation of EQUILIBRIUM-RANKING that isolates contributions from normalization of base models. We have additionally quantified the base rate of disagreement between generative and discriminative LM queries, helping explain why ER improves accuracy.

Please let us know if there are any other questions we can answer during the discussion period.

---

### Author Response · Authors · 2023-11-20
**Note from Authors**

Dear Reviewers,

As the discussion period is about to close, please let us know if there is any additional questions or anything else you would like to see in our revision. Thank you!

---

### Meta-Review · Area_Chair_sGL4 · 2023-12-06

**Metareview:**

The paper proposes a decoding strategy for language models based on a game theoretic formulation. The main objective is to reconcile the inconsistency between the model's response in generative and discriminative settings. The paper adapts recent regularized equilibrium approaches that operate directly on the outputs of samples from the generator and a discriminator to produce final answers. The method is a training-free method and consistently outperforms several baselines on a variety of QA tasks, including open-ended and math reasoning tasks. The novelty of the method and strength of the experimental evaluation justifies acceptance, and the paper has the potential to be very impactful. The main weakness is that the clarity of the presentation can be improved.

**Justification For Why Not Higher Score:**

The paper is likely to be impactful, but possibly not enough to justify oral.

**Justification For Why Not Lower Score:**

This will likely be one of the better papers at ICLR.

---

### Decision · Program_Chairs · 2024-01-16

Accept (spotlight)